# Characterization of Structural and Energetic Differences between Conformations of the SARS-CoV-2 Spike Protein

**DOI:** 10.3390/ma13235362

**Published:** 2020-11-26

**Authors:** Rodrigo A. Moreira, Horacio V. Guzman, Subramanian Boopathi, Joseph L. Baker, Adolfo B. Poma

**Affiliations:** 1Department of Biosystems and Soft Matter, Institute of Fundamental Technological Research, Polish Academy of Sciences, Pawińskiego 5B, 02-106 Warsaw, Poland; rams@ippt.pan.pl; 2Department of Theoretical Physics, Jožef Stefan Institute, Jamova 39, 1000 Ljubljana, Slovenia; horacio.guzman@ijs.si; 3Instituto de Ciencias Físicas, Universidad Nacional Autónoma de México, Cuernavaca 62210, Mexico; boopathialzheimer@outlook.com; 4Department of Chemistry, The College of New Jersey, 2000 Pennington Road, Ewing, NJ 08628, USA; bakerj@tcnj.edu

**Keywords:** SARS-CoV-2, spike protein, RBD, conformational space, structural stability, solvation energy, native contacts

## Abstract

The novel coronavirus disease 2019 (COVID-19) pandemic has disrupted modern societies and their economies. The resurgence in COVID-19 cases as part of the second wave is observed across Europe and the Americas. The scientific response has enabled a complete structural characterization of the Severe Acute Respiratory Syndrome—novel Coronavirus 2 (SARS-CoV-2). Among the most relevant proteins required by the novel coronavirus to facilitate the cell entry mechanism is the spike protein. This protein possesses a receptor-binding domain (RBD) that binds the cellular angiotensin-converting enzyme 2 (ACE2) and then triggers the fusion of viral and host cell membranes. In this regard, a comprehensive characterization of the structural stability of the spike protein is a crucial step to find new therapeutics to interrupt the process of recognition. On the other hand, it has been suggested that the participation of more than one RBD is a possible mechanism to enhance cell entry. Here, we discuss the protein structural stability based on the computational determination of the dynamic contact map and the energetic difference of the spike protein conformations via the mapping of the hydration free energy by the Poisson–Boltzmann method. We expect our result to foster the discussion of the number of RBD involved during recognition and the repurposing of new drugs to disable the recognition by discovering new hotspots for drug targets apart from the flexible loop in the RBD that binds the ACE2.

## 1. Introduction

Previous outbreaks of coronaviruses have threatened our modern societies [1,2]. However, neither of them stressed the worldwide health system and economy [3] more than the novel coronavirus. As of 30 October 2020, almost 45 million confirmed cases with a death toll over one million around the world have been reported. Thus, there is an urgent need to understand the molecular features of each of the proteins that are assembled into the virion. The fast spread of COVID-19 around the globe urges to devise viral deactivation strategies prior to cell recognition or block the viral replication mechanism, among others [4]. A key component in all coronavirus associated with cell entry is the spike protein (S) and, thus, its structural characterization is an essential step [5,6]. The typical spike protein is a homotrimer system and it plays a crucial role in the interaction between the virion and the host human cell membranes. The spike protein attaches itself to specific cellular receptors (i.e., human angiotensin-converting enzyme—ACE2) [7,8] and recently [9], it has been proposed that it can bind to nicotinic acetylcholine receptors (nAChRs) through the S protein. Such studies suggested that the S protein could favor binding to nAChR and avoid cell entry via the ACE2 receptor in the presence of nicotine. In the case of ACE2 receptor, the spike protein undergoes several conformational changes that engage different protein domains (e.g., receptor binding domain-(RBD), N-terminal domain-NTD, and S1 and S2 subunits) (see Figure 1). The first process that is believed to occur relates to the transition from down to up receptor-binding domain conformation. This transition prepares the virus for binding to the ACE2 receptor and the subsequent internalization of the virus through the formation of the endosome, later fusion of the viral and cell membranes, and the final release of the viral RNA into the cytoplasm [10]. It is clear that the spike protein is a crucial component in each aspect of the cell entry mechanism. Several studies [5,11,12,13,14,15] have elucidated how the novel coronavirus takes advantage of the spike protein structure to outperform SARS-CoV. For instance, the characterization of the mechanical stability of the RBD of Acute Respiratory Syndrome—novel Coronavirus 2 (SARS-CoV-2) has shown it to be stiffer (greater by 50 pN) compared to SARS-CoV [16]. This result has important consequences during binding to ACE2 (pre-fusion state) [17], as it can withstand Brownian and cellular forces and yet maintains close contact while priming of the spike protein by transmembrane protease serine 2 (TMPRSS2) occurs as part of S1 dissociation from S2 that enables the post-fusion mechanism [17,18]. In addition, in silico studies found a space correlation between the polybasic furin cleavage site Q_677_TNSPRRAR↓SV_687_ and surface residues located in the RBD region that recognize the ACE2 in SARS-CoV-2. Such effect was mediated by a long-range electrostatic interaction 10 nm apart [19]. Furthermore, the mutant D614G (a single residue change, D = aspartic acid by G = glycine) of the SARS-CoV-2 spike protein sequence, which became the dominant form globally at the end of March, displayed stronger transmissibility [20]. Also, this mutation correlated residues that are located about 7–10 nm from the SARS-CoV-2 RBD [19]. Certainly, it has also been suggested that those features in the spike protein could enhance its transmissibility and facilitate the post-fusion machinery in SARS-CoV-2 [21]. The intrinsic flexibility of the full ectodomain dictated by three hinges was characterized by cryo-electron microscopy (cryo-EM) and large-scale molecular dynamics (MD) simulation [22]. It shows the flexibility of the spike protein and the ability of the spike head to explore different orientations in space which allows it to scan the host cell surface in search of ACE receptors. A recent cryo-EM study has found a free fatty acid (FFA) pocket in each RBD [23]. The binding of the FFA linoleic acid stabilizes a locked conformation giving rise to reduced ACE2 interaction in vitro. The sugar coating of the surface in the spike protein by glycans are not only shielding to evade the immune system response as commonly believed [24], but also they may play a structural role by modulating the conformational dynamics of the spike’s RBD that is responsible for cell recognition [25].

In order to fight against COVID-19, several medical strategies have been developed (e.g., vaccines and monoclonal antibodies) [26,27,28]. New efforts in novel therapeutics employing short peptides, proteins, and natural resources, such as plant derivatives, are yet new ways for disabling the virion at the level of the spike protein [29,30,31,32]. In this regard, a stabilized form of spike protein with all RBD in down or one (or two) RBD in the up conformation is desirable for vaccine and therapeutic development because this conformation displays most of the neutralizing epitopes that can be targeted by antibodies to prevent cell entry [6,33]. These stabilized structures contain two consecutive proline substitutions in the S2 subunit in a turn between the central helix (CH) and heptad repeat 1 (HR1) that is important during the transition to a single, elongated α-helix in the post fusion conformation. Prefusion-stabilized spikes in closed and open states have been used to determine high-resolution spike structures by cryo-EM [5,7] that has been crucial for large-scale MD simulation restricted to a microsecond time scale. However, even with these substitutions, the SARS-CoV-2 ectodomain is unstable without the ACE2 receptor and typically difficult to express on mammalian cells, hampering biochemical research and search for novel vaccines. A recent single-molecule Förster Resonance Energy Transfer (smFRET) study characterizes the ensemble of conformations occurring in the spike which seems to be part of a dynamic equilibrium [34], but it does not lead to a quantitative assessment of the relative stability between conformers. A multiscale modeling that employs structure-based coarse-graining has shown the existence of a dynamic asymmetry that triggers the change in conformation of the closed to open states and the characteristic free-energy landscape shows the closed conformation as the ground state [34,35]. Here, we show an analysis of the relative stability considering dynamic contact map analysis. Our study shows the correlation between different conformations of the spike protein and its RBD, NTD, and S2 subunits and highlights the role of destabilization in order to get access to other conformations. This tool also allows determination of stable “hotspots” across the amino acid sequences and possible new targets for therapeutics. In addition, we obtain free energy differences between states of the spike protein without ACE2.

## 2. Materials and Methods

### 2.1. Modeling of the Full SARS-CoV-2 Spike Structures

The SARS-CoV-2 spike (S) protein conformations in the closed and open states were modeled with all RBD in down conformation and with one or two RBD in the up position. The different conformations studied correspond to three RBD in the down position (3down), one RBD in the up, and two RBD in the down positions (1up2down) and two RBD in the up and one RBD in the down position (2up1down). They were reconstructed based on cryo-EM structures with Protein Data Bank (PDB) codes 6VXX, 6VSB, and 6X2B, respectively. Several important loops in the spike protein were modeled based on the structure of the RBD bound with ACE2 receptor with PDB code 6M0J. The spike protein structures are fully completed. Same files for reconstruction have been used for each starting conformation. The wild-type (WT) sequences that describe the spike protein come from QIQ50172.1 stored in the GenBank database for SARS-CoV-2. The trimeric cryo-EM structures comprise mainly the sequence A27-S1147. In order to stabilize the cryoEM pre-fusion state several mutations were implemented, e.g.*,* P986K and P987V [5]. In structure files (PDB), however, there are missing residues that must be fulfilled to get the correct WT model, as well as, some residues that must be mutated to reconstruct the WT type. In particular, using as reference QIQ50172.1 starting at residue MET 1, residues ALA 570, THR 572, GLN 607, GLY 614, ARG 682, ARG 683, ARG 685, PHE 855, ASN 856, LYS 986, and VAL 987 were replaced from original PDB structures to reconstruct the WT sequence. The standard Needleman–Wunsch algorithm was used as implemented by Chimera visualization software to align the sequences and the missing loops were modeled by homology using MODELLER (version 9.25) [36]. The reconstructed configurations were optimized using standard energy minimization (1500 steps) and conjugate gradient (500 steps) algorithms available in MODELLER library. The disulfide bonds were the ones prescribed by the PDB files and 14 per single chain of the spike homotrimer. Our spike protein models were neither glycosylated nor cleaved in order to analyze the protein native contact for the global stability contribution.

### 2.2. All-Atom MD of the SARS-CoV-2 Spike and Its Conformations

Amber18 [37] was used to carry out all-atom simulations. The protein, water, and ions were all modeled using the FF14SB [38] and TIP3P [38,39] force fields. System energy was minimized using the CPU version of pmemd, while heating, equilibration, and production simulation stages used GPU pmemd. SARS-CoV-2 systems 3down, 1up2down, and 2up1down were placed into octahedral shells of TIP3P water of 14, 12, and 16 Å, respectively. Disulfide bonds (DBs) were added between cysteines which were close enough for a DB bond to form as defined by the starting model. The DBs were added using tLeap. The NaCl concentration for every simulation was 0.150 M NaCl. In order to use a 4 fs time step, hydrogen mass repartitioning was applied to the protein [40]. SHAKE was applied for hydrogen atoms, and an 8 Å real-space cutoff was also applied in the simulations. For long-range electrostatics we utilized PME with periodic boundary conditions. Minimization included 2000 iterations of the steepest descent method, subsequently 3000 iterations of the conjugate gradient minimization method. The heating protocol used: (1) increase of the temperature from 0 to 100 K (50 ps) in NVT, and (2) increase of the temperature from 100 to 300 K (over 100 ps) in NPT. During minimization and heating we used positional restraints of 10 kcal·mol^−1^·Å^−2^ for all C_α_ atoms. Subsequently, equilibration at 300 K (or equivalently 23 °C, as the room temperature) was carried out and the restraints on the C_α_ atoms were slowly removed, becoming reduced from a value of 10 kcal·mol^−1^·Å^−2^ to a value of 0.1 kcal·mol^−1^·Å^−2^ over 6 ns. For the production simulations, all restraints were removed and 200 and 320 ns production simulations were conducted for 3down and 1up2down SARS-CoV-2 conformations, respectively, and 100 ns for 2up1down case. Each of the production simulations started from the final coordinates at the end of the equilibration stage of simulation, and for each system we simulated five replicas. Pressure of 1 atm was maintained using the Monte Carlo barostat, and the system temperature of 300 K (or 23 °C) was held constant during production using the Langevin thermostat (a 1 ps^−1^ collision frequency was employed), using the native pressure and temperature control algorithm implemented in Amber18 [37]. For this work, in total 2.3 μs of all-atom MD data was used. A Zenodo repository is provided in which snapshots from the all-atom MD simulations can be obtained [41].

### 2.3. Differential Contact Map (dCM) Analysis

The contact map (CM) determination considers the Van der Waals (VdW) interaction between residues that are typically captured by a geometric based-approach denoted by extended overlap (OV) of the VdW spheres, which has been successfully used before to describe single proteins [42,43,44,45]. Furthermore, the chemical character of the residues can be also included as an additional part of the CM determination. The latter is denoted as the rCSU approach, which considers the chemical character of each atom, and respective possible bonds between two residues, into categories that count the number of stabilizing and destabilizing contacts per residue, defining a contact when both residues have a net stabilizing character. Together they form a robust CM methodology known as OV + rCSU contact map [46] that has been validated in the dynamic of large protein complexes [47,48,49,50,51]. This approach is used to get structural information from a specific geometry. This methodology is reliable enough to describe relatively small globular protein molecules. However, when applying it at a larger complex system, we may include contacts between residues that are not relevant to the system, e.g., contacts that belong to solvent-exposed flexible loops are less structurally relevant to describe a protein than the ones between residues of α-helices and β-strands. To get only the most structural relevant contacts, we can use the information available from the molecule’s dynamics. Computing the contact maps for each frame from a subset of MD frames, we can count how many times a given contact was identified. Then, a contact has a frequency (freq), defined by the number of frames where it was found over the total number of frames analyzed. This procedure gives a low frequency for contacts between flexible parts of the protein, while highly stable structures exhibit high frequency. This methodology, although simple, will require a large amount of sampling through a well-equilibrated MD trajectory.

Here, we use this methodology to sweep evenly distributed frames of the equilibrium MD trajectory of each system studied in this article to dynamically determine the high frequency contacts (freq > 0.9) between amino acids. In particular, from 10,000 frames of the closed conformation, we obtained a total number of 29,334 ± 82 contacts, 8000 frames for the 1up2down conformation showing 29,320 ± 741 contacts, as well as 8340 frames for the 2up1down case with 29,055 ± 718 contacts, which is a robust standard deviation of less than 3% from the total number of contacts. The source of contact fluctuations are essentially flexible loops that account for approximately 1772 amino acids based on secondary structural analysis, while helices and strands are approximately represented by 712 and 819 residues, respectively. The whole spike protein has a total of 3363 residues. Our objective is to discern between relevant and not relevant contacts, and a moving coil that eventually creates a contact is obviously less stable than a secondary structure. These small deviations in the number of contacts, and the consequent contacts frequencies, allow us to differentiate between relevant and not relevant contacts for the system’s structural stability, which is the main advantage of this methodology compared to a static analysis based on only one frame mostly taken from X-ray/NMR crystallography. To be able to compute such a large amount of contact maps, we implemented our own contact map software that implements the OV + rCSU approach, as detailed in Reference [46], and available via the Zenodo repository [41].

### 2.4. Poisson Boltzmann Calculations for the Spike Protein Energetic Characterization

The Poisson–Boltzmann method employed for the energetic characterization relies on the implicit solvent models, which averages the explicit influence of water molecules into a continuum dielectric description [52]. Consequently, a dissolved molecule is expressed as a multi-dielectric infinite domain within our scheme containing two regions: the proteins (solute) and water (solvent). The protein encapsulating boundary between the solute and solvent is given by the Solvent Excluded Surface (SES). In the case of the spike proteins, the approach is divided in two regions:Coronavirus spike protein structures (Ω_1_).External (Ω_2_), representing the solver.

Every single region is described by its dielectric constant (ε_1_, ε_2_), and salinity (*C*_1_, *C*_2_), which is considered as zero in the solute. The solvent parameters are appropriate proper for a solvent in the external region (Ω_2_) [53]. The protein partial charges are modeled as static point charges at the respective atomistic coordinates, while the salt ions in the solvent are treated as mobile point charges that allocate according to Boltzmann statistics. Introducing continuum electrostatic theory on this charge allocation leads to a system of coupled partial differential equations:∇^2^*ϕ*_1_ = ∑*_k_**q_k_δ*(*r_k_*)
(∇^2^ − *κ*_2_^2^) *ϕ*_2_ = 0*ϕ*_1_ = *ϕ*_2_*∈*_1_(∂*ϕ*_1_/∂*n*) = *∈*_2_(∂*ϕ*_2_/∂*n*)(1)

In these equations, the electrostatic potential is *ϕ* the inverse of the Debye length is given by *κ* (which depends on ionic strength *C*), *ε* is the dielectric constant, and the partial charges *q* of the protein. Γ_1_ is the SES on the outside of the Spike protein, interfacing Ω_1_ with Ω_2_. The unit vector *n* is normal to the SES, and points away from the region enclosed by the surface. The solvation energy is one of the quantities of interest, defined as the work required to bring the protein from vacuum into the solvent. The charges inside the protein are considered as Dirac delta functions, letting the solvation energy as:Δ*G_solv_*  =  ∫*_Ω_*_1_*ρϕ_reac_*  =  ∑*_k_**q_k_ϕ_reac_*(*r_k_*)(2)
where *ϕ_reac_*  =  *ϕ* − *ϕ_coul_* is defined as the reaction potential, *ρ* is the charge distribution, and *r_k_* charges locations. A second energetic source in the point-charge distribution of the partial charges in the protein which comes from a Coulomb type energy (see Equation (4)). Then, the total electrostatic contribution to free energy is the sum of the solvation energy and Coulomb energy:*G_elec_*  =  Δ*G_solv_* + *G_coul_*(3)
where:*G_coul_* = (½) *∈*_1_∑*_k_*∑*_j_**q_k_*(*q_j_*/4*π*|*r_k_* − *r_j_*|)(4)

Here, we choose PyGBE [52] as the Poisson-Boltzmann solver to efficiently compute the electrostatic potential and energy in Equations (1) and (3). The system of partial differential equations are formulated in the form of boundary integrals, required to mesh the solvent excluded surface, and solves the resulting system of equations is solved with a boundary element method [54]. From a viewpoint of MD trajectories, the initial structures for Poisson–Boltzmann (PB) analysis corresponded to the final snapshot of the MD trajectories. Here we employ three different spike protein conformations. In addition to the structural positions the charges and vdW radii are also required by the PB solver to calculate solvation and Coulomb energies, mostly given in a pqr format. Here, we remark that PB-energies provide a static picture of the solvation and Coulomb energies at a certain point of the MD simulations trajectory. In this regard, PB methodology requires well-equilibrated molecular trajectories with atomistic resolution [41].

## 3. Results and Discussion

### 3.1. Relative Stability Analysis of the SARS-CoV-2 Spike Protein Conformation

The characterization of the full space of conformations of the spike protein and the intrinsic stability by MD dynamics simulation is still limited by the length and time scales needed to sample the conformational space of the spike protein. In a recent study carried out by smFRET [34], the closed and open conformations were sampled by tagging certain amino acids in the NTD and RBD regions and monitoring their positions in space. In particular, this methodology allows the reconstruction of a restricted free energy landscape that shows a high stability for the closed state over the open, as it is believed in the absence of the ACE2 receptor. However, it does not highlight the amino acid regions or the hotspots of stability which can be used for therapeutics by developing small drugs that bind those regions and disrupt the stability associated with them. In MD simulation the transition from closed to open states is only possible via large forces acting on the RBD in a nonequilibrium fashion. This protocol captures the transitions, but it is only thermodynamically consistent in the limit of a large number of pathways through the Jarzynski equality [52]. In this section, we define the local stability of the homotrimer spike protein defined by the difference of the number of native contacts associated with single amino acids between two conformations. The change of the number of contacts under a conformational change shows a measure of the loss or gain of the local stability. We test our dCM methodology on three different conformations of the spike protein. A graphical representation of our findings is presented in Figure 2 and Figure 3. In those figures, panel A highlights the position of the destabilizing residues in the spike protein, panel B depicts the same for the stabilizing residues, and panel C shows the contact map representation. Such graphical analysis shows the tendency to stabilize the intra protein interactions at the cost of destabilizing the interchain contacts which seems to indicate that a single RBD down to up transition affects the overall stability and could facilitate shedding of S2 from S1 once a surface proteases cleave the S1/S2 position. Based on our analysis, the 1up2down conformation shows a smaller number of destabilizing residues than the 2up1down conformation. However, the stabilizing residues per protomer in up conformation seem to be comparable. In addition, we reported in Appendix A the chemical character of the high-frequency contacts. We noticed that hydrophobic interactions are generally larger in number over the polar–polar and electrostatic interactions in the destabilizing regions. Here, we prove that among the two studied conformations of the open state the most stable is the 1up2down system. In Table 1, we reported the number of the destabilizing and stabilizing residues and their corresponding high-frequency (freq > 0.9) contacts. At first, we observed the large number of destabilizing residues, 92 and 502, which account for 3% and 15% of the total system for 3down to 1up2down and 3down to 2up1down, respectively. This is an indication that a transition from closed state to 1up2down implies the destabilization of a small number of native contacts. However, such a type of transition is more serious as it destabilizes several parts of the protein once it exposes 2 RBD in up conformation. Note that a small gain in stability is reported in very key places of the spike protein such as flexible loops that are engaged in cell recognition at the RBD (chain B) and stabilized by formation of new intrachain contacts. In the absence of ACE2, our study renders the open state less stable than the closed state which agrees with the smFRET study [34] that identifies the closed state as the most populated state in the lack of ACE2. Moreover, our study shows an additional structural stability associated with almost twice more high-frequency contacts in 1up12own than 2up1down conformations (see Table 1).

### 3.2. Insight into the RBD SARS-CoV-2 Conformations

In Figure 4A, we show the RBD in chain B in contact with the RBD in chain C and NTD in chain A in a closed conformation. Based on the analysis of the high-frequency contacts (freq > 0.9) we find out 25 stabilizing residues in down conformation making 83 contacts that are key for the 1up2down case forming 40 contacts (see Figure 4B). Such conformational transition emerges through the rearrangement of certain residues located at the bottom of the RBD, close to the hinge region (H519-S530) in chain B that change conformation and breaks 10 high-frequency contacts between RBD in chain B and S2 in chain A mediated by the hinge region (see Figure 1) in order to transit to the new position. The list of those 10 contacts is given in Appendix A. By considering a range of energy values for native contacts in protein, i.e., 1.0–1.5 kcal/mol. The energy associated with the hinge regions in the 3down to 1up2down transition will be in the range of 10–15 kcal/mol. The identification of the residues that create contacts to stabilize a conformation is of great importance for novel therapeutic methods that can target binding to enhance or disrupt the established contacts for a given purpose. For instance, the work by Toelzer, C. et al. [22] has unveiled an FFA binding a hydrophobic pocket that locks the structure of the spike in the closed state (see Figure 4C). Such a process is achieved by establishing additional contacts between two nearest RBDs through the FFA. Indeed, such a process is possible due to few pre-existing native contacts in the vicinity of those two RBDs. Those contacts engage the RBD segment (R408-Q409) in chain B and a gating helix (Y365-Y369) in the other RBD in chain A. The latter motif is crucial for the transition from apo conformation to FFA conjugated with protein that has been reported to lock the RBD in the closed state. Our analysis shows the presence of other stabilizing contacts among the gating helix in a given chain and the next chain namely, Q409-Y369, Q414-Y369, T415-Y369, G416-Y369, K417-N370, K417-Y369, and D420-Y369. Due to the symmetry in the closed state, the set of contacts are similarly distributed among all chains. A recent study by Casalino et al. [24] has elucidated the stabilization of the open conformation by two additional N-glycans at positions N165 and N234, that are located at the NTD in chain A and according to all-atom MD simulations they modulate the RBD conformational dynamics. Our analysis further shows the stabilization of the RBD in chain B in up conformation through residues located at the NTD in chain A, which are close to N165. Here, we get T167(A)-R357(B), F168(A)-R357(B), and E169(A)-R357(B), and close to N234 we have P230(A)-T523(B), P230(A)-A520(B), and P230(A)-P521(B). In addition, we also found few residues in RBD in chain A interacting with the RBD in chain C, both considered in down conformation. Those contacts are K417(A)-Y369(C), R403(A)-A372(C), T415(A)-P384(C), K417(A)-N370(C), and L455(A)-N370(C) and more interestingly, we observed an interaction between RBD in the up conformation and the closer RBD in the down position by establishing contacts such as F377(B)-F486(C), T385(B)-K458(C), S375(B)-F486(C), S383(B)-F456(C), T376(B)-F486(C), T385(B)-F456(C), and F374(B)-F486(C).

### 3.3. Energetic Calculations of the Spike Protein Conformations

The role of electrostatic interactions in the spike proteins have been calculated by the well-known Poisson–Boltzmann (PB) method [53]. It is composed by two contributions, namely, the solvation energies and a complementary term originated by coulomb interactions. The solvation energies which have been calculated by capturing the spike proteins structure from the MD trajectory [41], and they depend on the solvent-excluded surface (SES) and the partial charges of the solute. A second contribution arises from the point charges distribution in the solute, given by the Coulomb energy, (see the Section 2: Materials and Methods for the theoretical background). The PB scheme has several advantages, when modelling very large protein assemblies such as the case of viral capsids and structural viral subunit of the coronavirus (e.g., spike proteins). The PB scheme is a very optimized technique that fits into parallel computing and allows the calculation of the solvation free energy for a single MD snapshot on the time scale of an hour for systems consisting of about 100,000 atoms. [53].

In Table 2, we have calculated the PB-energy differences between the spike proteins with at least one RBD up and with the closed spike configuration as a reference. Assuming physiological conditions given by a pH of 7 and ionic-strength of 150 mM, we observed that the ground energy state is the spike conformation with 3down. In addition, the most favorable open conformation is given by the 1up2down conformation, followed by the 2up1down conformation. Interestingly the energy barrier to reach a 2up1down conformation seems to have an energetic penalty much higher compared to as the first transition, i.e., 3down to 1up2down. Here, we remark that PB-energies provide a static picture of the solvation and Coulomb energies at a certain point of the MD simulations trajectory. Table 2 shows the calculation of the ΔG of energy for 3 MD snapshots corresponding to a tuned mesh refinement (4.16 elements/Å^2^) set of the three system-ending conformations [41].

The PB energy calculation can be considered as an initial overall energetic comparison between the different spike protein conformations, which also goes in line with the contact map stability analysis. However, given the relevance of the recognition process, we are not providing a final assessment of the ΔGs. In fact, we suggest extending this study and comparing results between different free-energy calculation methods [54] applied to the spike protein including a reconstruction of the envelope membrane. The inclusion of a reconstructed membrane may provide further insight into the large and flexible conformational space of the spike proteins in a more robust and accurate manner.

## 4. Conclusions

In the current COVID-19 pandemic, the development of novel diagnostics and antiviral therapies is of high priority. We expect that our structural and energetic studies can provide additional information regarding the space of interactions mapped by certain key residues that are crucial for stabilization of the spike protein during an apparent dynamic equilibrium mediated by transitions from close to open conformations prior to ACE2 recognition. In particular, we found several high-frequency contacts formed between the NTD (in chain A) and RBD (in chain B) that are responsible for the local conformational stability, and our data suggest that these high-frequency contacts play a role during the transition from closed to open state. Additional studies using lifetime of contacts can be used to study stability of relevant residues responsible for ACE2 recognition. Our analysis in the absence of ACE2 shows that this transition occurs at the energetic cost of breaking very high-frequency contacts between the RBD hinge and the S2 region in chain B and A, respectively. The rearrangement of those residues has an energetic cost in the range of 10–15 kcal/mol which is consistent with the PB analysis that quantifies the change in free energy on the order of 10.4 kcal/mol for 3down to 1up2down. Our studies also show the large energetic cost required to transit from closed to 2up1down conformation (~30 kcal/mol) in the absence of the ACE2 receptor which can be associated with mechanical loading and virus-cell collisions at the early stage. This result indicates the propensity of the spike protein to likely be found in the 1up2down conformation prior to interacting with the cell surface. Further studies based on single molecule force spectroscopy can help to differentiate spike protein conformation according to their mechanical properties (e.g., Young’s modulus).

In perspective, we aim to provide enough information about possible target sites to destabilize the spike conformations of the closed and open state (i.e., 1up2down or 2up1down). One possibility is that our work can be combined with other studies on the binding of natural compounds derived from plant sources [55]. In this regard, several herbal derivatives having hepatoprotective, anxiolytic, antidepressant, nootropic, antimicrobial, anti-inflammatory, antioxidant, anti-stress, anticonvulsant, cardio-protective, antitumor, anti-genotoxic, anti-Parkinson, and immunomodulatory properties can be used to target inhibition of spike protein conformation. Recently, it has been approved to use them as the alternative antiviral inhibitor for COVID-19 patient treatment [28,29]. In view of the above considerations, in the future, we plan to investigate and determine the efficacy of the potential herbal candidates along with FDA approved drugs (e.g., Remdesivir, Lopinavir, Favipiravir, and Hydroxychloroquine) [4] against SARS-CoV-2 spike protein by weakening its RBD interactions with ACE2. This can be accomplished by combining in-silico docking and molecular dynamics simulation in order to determine the destabilizing residues and how their contacts with the RBD are broken upon ligand binding. From this work, we will be able to propose possible computationally promising compounds that can be further probed experimentally.

## Figures and Tables

**Figure 1 materials-13-05362-f001:**
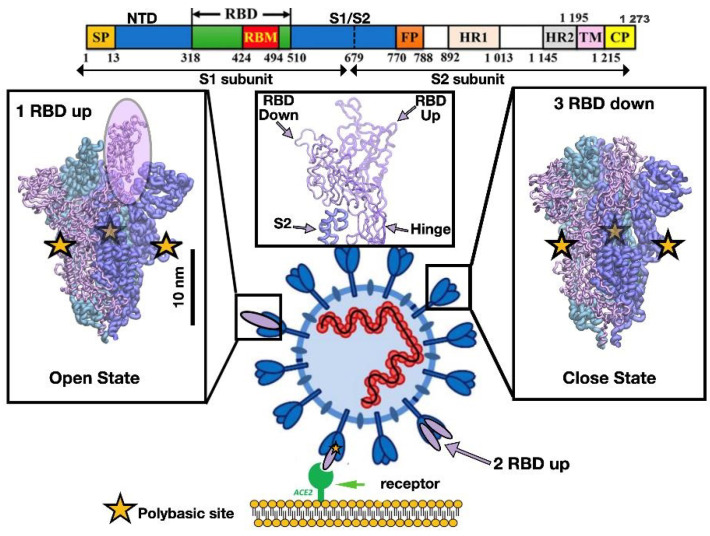
Representation of the different conformations of the receptor-binding domain (RBD) in the Acute Respiratory Syndrome—novel Coronavirus 2 (SARS-CoV-2) spike protein. Cell recognition is initiated by the RBD transition from down to up conformation that involves the RBD detachment from S2 mediated by the hinge region as depicted in the middle panel. Then, as a result of the high affinity between the RBD in the up conformation and the angiotensin-converting enzyme 2 (ACE2) receptor, binding takes place. The sequence of one chain of the spike protein is shown on top as well as the residue numbers for several protein domains. The bar shows the typical length scale for the whole system. The fusion of the viral and cell membrane takes place by surface proteases that cleave each chain at the polybasic sites (yellow stars) located at the interface of S1/S2 subunits.

**Figure 2 materials-13-05362-f002:**
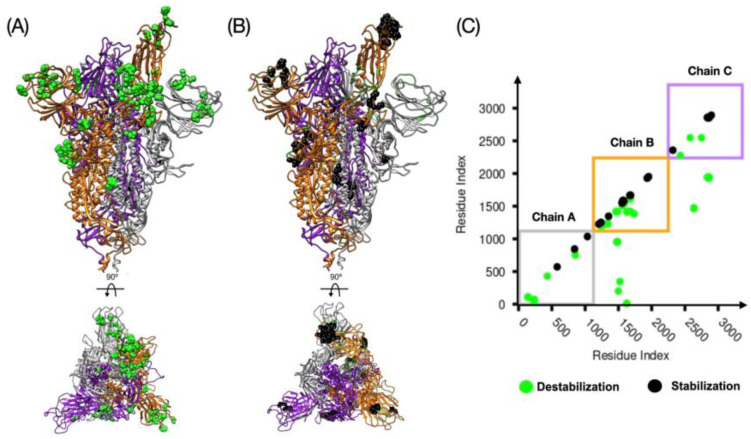
The differential contact map (dCM) analysis (f > 0.9) of the spike protein with 1up2down RBD. The secondary structure of the chain A–C are depicted in gray, yellow, and purple, respectively. Panel (**A**) shows the structure of the homotrimer and in green the amino acid residues that destabilize the spike protein in this conformation. The bottom of panel (**A**) shows the same structure rotated by 90° as indicated, showing that most of the destabilizing residues are positioned between the RBD (in up position) from chain B and the RBD and NTD from chain A. Panel (**B**) shows the stabilizing residues in black. Panel (**C**) shows the plot of the contact map. The squares follow the color convention for each protomer and each dot represents a high-frequency contact (i.e., green = destabilizing and black = stabilizing).

**Figure 3 materials-13-05362-f003:**
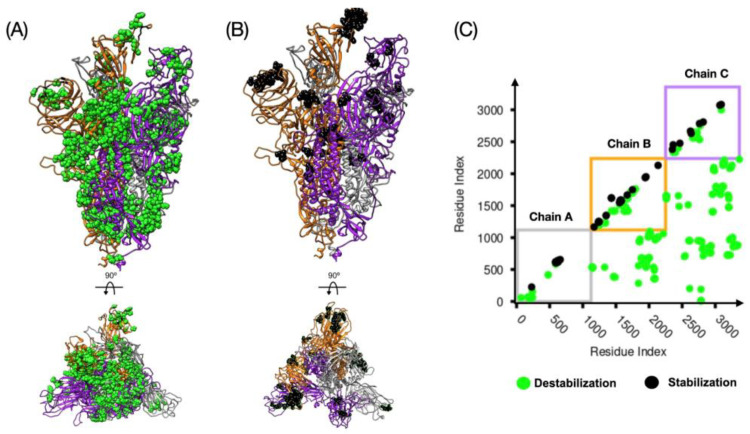
The same analysis as in Figure 2 but for the spike protein with 2up1down RBD conformation. Panel (**A**) shows the structure of the S protein and in green the amino acid residues that destabilize the spike protein in the 2up1down conformation. The bottom of panel (**A**) shows the same structure rotated by 90° as indicated, showing that most of the destabilizing residues are positioned along S1, RBDs and NTDs in chains B and C. Panel (**B**) shows the stabilizing residues in black. Panel (**C**) shows the plot of the contact map. The squares follow the color convention for each protomer and each dot represents a high-frequency contact (i.e., green = destabilizing and black = stabilizing).

**Figure 4 materials-13-05362-f004:**
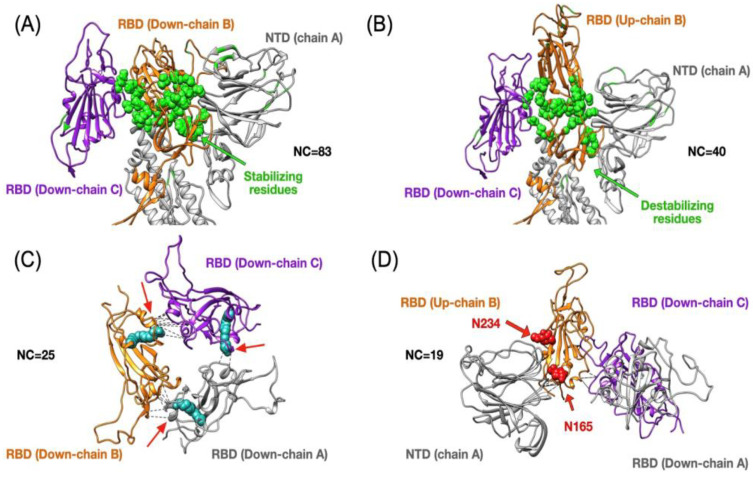
Top panels show changes in terms of stability between the closed (3down) and open (1up2down) states. Panel (**A**) shows the case of the RBD in chain B stabilized by neighboring protein domains such as the RBD and NTD from chain C and A, respectively. The stabilizing 25 residues are highlighted in green and they establish several high-frequency native contacts (NC) equal to 83. Panel (**B**) shows the same set of residues from panel (**A**) that are destabilized in the open state forming 40 contacts. Panel (**C**) shows the stabilization due to 25 contacts between all RBDs in the closed state. The structure of the FFA (in cyan) has been superimposed onto our results as it was shown to lock the closed state by forming contacts between two adjacent RBDs. Panel (**D**) depicts the RBD in up conformation that has been stabilized by 19 contacts formed between the NTD and two other RBDs. The positions of two N-glycans that assist structurally by making extensive interaction with the RBD in the up conformation have been superimposed onto our structure. We highlight the residue contacts that are responsible for stabilization by dashed black lines.

**Table 1 materials-13-05362-t001:** Differential contact map analysis of the spike protein calculated for every single protomer A, B, and C. We show the total number of destabilizing (denoted in bold) and stabilizing residues and their corresponding number of intrachain contacts with freq > 0.9 in parentheses.

Spike	dCM	Chain A		Chain B		Chain C	
1up2down	3down	**24 (9)**	6(3)	**54 (26)**	32 (24)	**14 (3)**	14 (11)
2up1down	3down	**156 (18)**	18(9)	**196 (61)**	47 (32)	**150 (10)**	17 (13)
2up1down	1up2down	**122 (15)**	12(6)	**111 (18)**	6 (3)	**110 (2)**	4 (2)

**Table 2 materials-13-05362-t002:** ΔG calculated by the Poisson–Boltzmann (PB) method at a pH of 7 and an ionic-strength of 150 mM.

Spike Conformation “C”	Spike Conformation Reference “R”	Δ(C-R) PB Energies (kcal/mol)
1up2down	3down	10.4
2up1down	3down	32.5

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
