# Peer review of "Characterization of Structural and Energetic Differences between Conformations of the SARS-CoV-2 Spike Protein"

_materials, 2020, doi:10.3390/ma13235362_

Round 1
Reviewer 1 Report
Authors of the study "Characterization of Structural and Energetic Differences Between Conformations of the SARS-CoV-2 Spike Protein" provide an interesting and useful approach at tackling the dynamics of SARS spike protein in particular, and multidomain proteins in general. The study is indeed very timely and could have potential impact on predictions of ACE2 binding inhibtors that could act by locking the RBD in the closed conformation.
While the manuscript reads well, there are some issues that should be addressed before I could recommend the publication:
1. While I definitely appreciate Author's work to collect all the interesting background details on the structure and function of the spike protein, I do not think this information is pertinent to understanding of the results and could be better suited for a dedicated review paper. For the ease of reading I would recommend shortening it to only contain details directly relevant to the study.
2. Methods section does not provide enough details. For instance, what protonation states were assigned to the residues? These could alter the dynamics and would certainly influence the PB analysis. On a similar note - was the capping of the terminal residues used? Especially N-terminus, if charged, could alter PB calculations. Some more detail on the minimization of the modeller-derived loops should be added, "few steps of energy minimization algorithms" sounds a bit vague. What is the frequency of writing the coordinates? Was the furin site cleaved?
3. PB analysis
From the Results it seems that only one snapshot was used for this calculation. It seems strange, since Authors emphasize just few lines above the advantages of the PB method in rapid calculation of the solvation free energies for large protein complexes. I understand the Authors plan a separate study in that topic, but to make the PB calculatin in the present study more usefull to the community, the Authors should expand this section. To eliminate the possibility that either of the system has reached some unlikely conformation when the simulation was stopped, DeltaG calculation should be provided for more snapshots, either uniformly spanning the trajectory, being representative structures from a clustering analysis or perhaps selected from the dynamic contact map analysis as the structures with the most/least contacts? As a minimum, provided five replicas were used, Authors should compare values for final frames from each of the replicas.
4. Single particle structures of the SARS all suffer from poor resolution in the loop regions of the NTD and partially RBD. Modelling these loops inevitably leads to uncertainties. I imagine the way the loops are reconstructed, if different between the systems, could affect very differently the (de)stabilising contacts. Were the loops modeled in a consistent way between the systems? How would the contact map analysis results change if only contacts between resolved parts of the structures would be considered?
5. Glycosylation
As Authors note, glycans thoroughly cover the surface of the spike, including RBD and NTD domains. Authors did not decide to glycosylate the system (which should be stated in the Methods section), yet some discussion of what the role of the glycans in the (de)stabilising contacts would be welcome. Are the contacts involving the sequons of high or low frequency? Are they involved in the de-stabilising contacts? Authors should comment on that and extend the discussion beyond the N165+N234. Perhaps re-introduction of the two GlcNac residues (typically visible in the EM maps) to the analysed frames could shed some additional light on that?
6.The dramatic increase in the destabilising contacts of the 2up1down conformation far away from the RBD/NTD domains is very interesting. 6vsb and 6x2b are remarkably similar in this region. Could the Authors speculate in the Conclusions on the mechanism of how the opening of the RBDs propagates to these far regions? It can be of high interest in the context of the spike conformational change to the post-fusion state, e.g. is binding of a second ACE2 and thus stabilising 2up the critical step to trigger the refolding?
7. The Results section would benefit from a thorough check. There are some sentences
that are very difficult to follow or contain some unclear sections, e.g. "when tackling large biggish protein assemblies". The abbreviation aa was not defined.
8. The abbreviation of LA is not used, whereas the abbreviation of the FFA could be introduced on page 2 to avoid confusion of LA-FFA when reading through the Fig 4.
Finally, I would like to complement the addition of the Zenodo repository to the manuscript! Just a small comment - it would be even more useful to the community if the Authors could also provide the input files for the production runs. The PDB files should have the chains annotated, otherwise it is difficult to directly compare the results which use the "chain A,B,C" terminology.
Author Response
Reviewer #1
Authors of the study "Characterization of Structural and Energetic Differences Between Conformations of the SARS-CoV-2 Spike Protein" provide an interesting and useful approach at tackling the dynamics of SARS spike protein in particular, and multidomain proteins in general. The study is indeed very timely and could have potential impact on predictions of ACE2 binding inhibitors that could act by locking the RBD in the closed conformation.
While the manuscript reads well, there are some issues that should be addressed before I could recommend the publication:
- While I definitely appreciate Author's work to collect all the interesting background details on the structure and function of the spike protein, I do not think this information is pertinent to understanding of the results and could be better suited for a dedicated review paper. For the ease of reading I would recommend shortening it to only contain details directly relevant to the study.
Answer: We have revised our MS.
- The Methods section does not provide enough details. For instance, what protonation states were assigned to the residues? These could alter the dynamics and would certainly influence the PB analysis. On a similar note - was the capping of the terminal residues used? Especially N-terminus, if charged, could alter PB calculations. Some more detail on the minimization of the modeller-derived loops should be added, "few steps of energy minimization algorithms" sounds a bit vague. What is the frequency of writing the coordinates? Was the furin site cleaved?
Answer: We did not titrate any amino acid to a non-standard protonation state. If we had done this, the information would have been included in the Methods section, so there is no missing information from the Methods in this regard. Furthermore, if one truly wants to investigate the influence of specific protonation states on a protein’s dynamics, and there are certain amino acids in mind that are considered important, then a constant pH replica exchange simulation for example would be the way to proceed and not the simulation of discrete protonated states of the system, especially one as large as the coronavirus spike. This is outside of the scope of this study, which targeted a standard state of the spike protein. The N-terminus and C-terminus of each protein were not capped, which is the default setting in AMBER. Similar to the protonation state response, if we had used any special capping in the procedure this would have been reported in the Methods section. As far as the authors are aware, it is not common practice in the MD literature to report when the termini are not capped, so the absence of that detail in the Methods implies that capping was not performed in this study. The default in AMBER is set to not use caps since in a cellular environment generally proteins are not capped. The reviewer is correct that those charges could interact with other regions of the protein, but this is true of any charge, and there is nothing special about the N-terminus and C-terminus charges compared to any other charge in the system. If there is a particular reason to cap the N- or C-terminus to compare with a specific experimental study, that of course could be done, however it is not the purpose of this work. Coordinates are written every 10 ps, which is standard practice for systems as large as the one that is being studied here. The initial structures of all missing sequences were recreated from available PDB structures. Then, the initial reconstructed configurations were checked for clashes, and the identified residues were energy minimized using standard energy minimization (1500 steps) and conjugate gradient algorithms (500 steps) available on MODELLER library. That procedure was repeated as many times as necessary to remove all clashes from the select residues. The resulting configuration undergoes an initial energy minimization using GROMACS and CHARMM27 force-field to be prepared for pre-equilibration and MD simulations. The furin site was not cleaved in our simulation as it highlights the problem of global spike stability. Additional information has been included in line 153-158.
- PB analysis
From the Results it seems that only one snapshot was used for this calculation. It seems strange, since Authors emphasize just a few lines above the advantages of the PB method in rapid calculation of the solvation free energies for large protein complexes. I understand the Authors plan a separate study in that topic, but to make the PB calculation in the present study more useful to the community, the Authors should expand this section. To eliminate the possibility that either of the system has reached some unlikely conformation when the simulation was stopped, DeltaG calculation should be provided for more snapshots, either uniformly spanning the trajectory, being representative structures from a clustering analysis or perhaps selected from the dynamic contact map analysis as the structures with the most/least contacts? As a minimum, provided five replicas were used, Authors should compare values for final frames from each of the replicas.
Answer: We appreciate this comment and have clarified in the description of the PB computational efficiency the level of optimization (line 398). Furthermore, our G values represent an initial assessment (as declared in the MS) on well-relaxed MD structures for each conformation. A more comprehensive study will be carried out in the future.
- Single particle structures of the SARS all suffer from poor resolution in the loop regions of the NTD and partially RBD. Modelling these loops inevitably leads to uncertainties. I imagine the way the loops are reconstructed, if different between the systems, could affect very differently the (de)stabilising contacts. Were the loops modeled in a consistent way between the systems? How would the contact map analysis results change if only contacts between resolved parts of the structures would be considered?
Answer: We thank the reviewer for this insightful comment. During the loop reconstruction we tried to be the most systematic possible. The loops were reconstructed following a regular modelling protocol. First we gather Information available in other PDB files and then we use the same information on the different spike protein conformations in order to reconstruct the missing structural gaps (i.e. loops). Of course, each conformation (close and open) belongs to different cryo-EM experiments, so it is expected minor changes after reconstruction and equilibration of those regions. In the original spike protein PDB files, the loop that contacts the ACE2 is not present and for being a key component in cellular recognition, then a need for reconstruction. It has been reported that in the absence of those loops the system drifts away from the equilibrium towards a new conformation (i.e. shedding of S1 from S2, postfusion) which does not maintain the pre-fusion (open state) conformation of the current study. So, we expect in the absence of those regions the pre-fusion state will not be well-defined. A statement in the MS has been included about it (lines 143-144).
- Glycosylation
As Authors note, glycans thoroughly cover the surface of the spike, including RBD and NTD domains. Authors did not decide to glycosylated the system (which should be stated in the Methods section), yet some discussion of what the role of the glycans in the (de)stabilising contacts would be welcome. Are the contacts involving the sequence of high or low frequency? Are they involved in the de-stabilising contacts? Authors should comment on that and extend the discussion beyond the N165+N234. Perhaps re-introduction of the two GlcNac residues (typically visible in the EM maps) to the analysed frames could shed some additional light on that?
Answer: We appreciate this comment. Although the glycosylation state has been acknowledged to play not only the protective role in order to evade the immune system, it has recently shown by Casalino et al. [23] a role in the stabilization of the RBD in the up conformation. However, our aim is not to validate such results, but to show the relevance of the high-frequency (protein) contacts which are energetically and contribute to global protein stability. As a perspective, we mention the possibility to study contact between N-glycans and residues in a more dynamic manner in order to infer other regions of stabilization due to a glycan-protein contribution. We have stated in the MS that the spike protein was not glycosylated (lines 157-158). Additional discussion beyond the N165 and N234 will be explored in the future.
6.The dramatic increase in the destabilising contacts of the 2up1down conformation far away from the RBD/NTD domains is very interesting. 6vsb and 6x2b are remarkably similar in this region. Could the Authors speculate in the Conclusions on the mechanism of how the opening of the RBDs propagates to these far regions? It can be of high interest in the context of the spike conformational change to the post-fusion state, e.g. is binding of a second ACE2 and thus stabilising 2up the critical step to trigger the refolding?
Answer: We thank the reviewer for his/her comment. We consider it as an interesting question, but we refrain from speculation at this stage of the work. Certainly, the mechanism of destabilisation of domains away from the RBD/NTD under spike conformational change will represent a target of another study on allosteric communication pathways in the future.
- The Results section would benefit from a thorough check. There are some sentences
that are very difficult to follow or contain some unclear sections, e.g. "when tackling large biggish protein assemblies". The abbreviation aa was not defined.
Answer: We have reviewed this section.
- The abbreviation of LA is not used, whereas the abbreviation of the FFA could be introduced on page 2 to avoid confusion of LA-FFA when reading through the Fig 4.
Answer: We have revised the MS.
Finally, I would like to complement the addition of the Zenodo repository to the manuscript! Just a small comment - it would be even more useful to the community if the Authors could also provide the input files for the production runs. The PDB files should have the chains annotated, otherwise it is difficult to directly compare the results which use the "chain A,B,C" terminology.
Answer: We have done our best to include all relevant information for reproducibility of our results in the Zenodo, including the labelling of chains used in this paper.
Reviewer 2 Report
The authors studied the structural stability of spike protein trimer, which involved in the novel coronavirus disease COVID-19, by simulating three RBD varieties: 3down, 1up2down and 2up1down. These conformations cover both the closed and open states of the spike protein. Their contact map analysis has shown that the switch between these conformations has a energetic cost of 10-15 kcal/mol. The topic itself is very interesting, but several aspects are unclear in the manuscript.
1) In Table 1 and Figs.2-3, the number of contacts and their frequency have been counted. Can the authors classify these contacts into different groups, like hydrogen-bonds, electrostatic interactions, hydrophobic interactions, et.al. This will give the readers a feeling that which is the dominant interactions between the interacting partners.
2) Comparing with the frequency of the contacts, perhaps the life time of them is more important address the structural stability of the spike protein.
3) There are many published works on the topic of "COVID-19". In the paper "How Does the Novel Coronavirus Interact with the Human ACE2 Enzyme? A Thermodynamic Answer", it is mentioned that compared with the old RBD, the Gibbs free energy of binding to hACE2 enzyme is increases from 5.11 to 8.39 kcal/mol for the new RBD. Why the free energy switch between closed and open state is 10-15 kcal/mol in this work?
4) In the same work as mentioned in point 3), they authors have detected many active sides of forming hydrogen bonds. Are the obtained contacts match with these active sides?
5) It is known that protein structures are also dependent on the chosen all-atom force field. Therefore, the contact map may be also sensitive to the choice of the force field. Can the authors verify their force field choice? At least the criteria should be mentioned.
6) Typos. For example, "as a pandemic..As" in page 2, "Delta Gs" instead of "\Delta Gs" in page 11. The authors use "Severe Acute Respiratory Syndrome" at one place but "severe acute respiratory syndrome" at another. Please use the same format.
Author Response
Reviewer 1
The authors studied the structural stability of spike protein trimer, which is involved in the novel coronavirus disease COVID-19, by simulating three RBD varieties: 3down, 1up2down and 2up1down. These conformations cover both the closed and open states of the spike protein. Their contact map analysis has shown that the switch between these conformations has an energetic cost of 10-15 kcal/mol. The topic itself is very interesting, but several aspects are unclear in the manuscript.
1) In Table 1 and Figs.2-3, the number of contacts and their frequency have been counted. Can the authors classify these contacts into different groups, like hydrogen-bonds, electrostatic interactions, hydrophobic interactions, et.al. This will give the readers a feeling which is the dominant interactions between the interacting partners.
Answer: We appreciate the insightful comment and we have added a very detailed Table S1 in the supplementary material. Regarding the analysis of the dominant interactions we can say that hydrogen bonds possibly mediated by polar residues are more dominant than hydrophobic and electrostatic interactions in the destabilized conformations. This result is supported by the larger number of polar residues in the destabilized structures. Additional information in track-change mode is given in lines (338-341) and in the SM.
2) Comparing the frequency of the contacts, perhaps the lifetime of them is more important to address the structural stability of the spike protein.
Answer: We thank the reviewer for the opportunity to expand on the meaning of the frequency of contacts in our study. The definition of the overall structural stability that is based on the high-frequency (or strength) of protein contacts is a probabilistic quantity that allows a systematic comparison among different spike protein conformations that is our aim in this study. On the other hand, the lifetime of a single contact is the measure of the longest continuous period of this contact before it breaks (or discontinues) between consecutive MD frames saved every Δτ (i.e. in the range of few fs). Although the lifetime can be calculated for each protein contact, this number will be subjected to a good choice of Δτ. In this regard, the comparison per residue between two spike conformations is simple through the frequency definition which only depends on the length of the sampling. The meaning of the high-frequency contacts with freq > 0.9 indicates that more than 90% of the time a given contact is found. In addition, our data was saved every 0.5 ps, due to the large system size, which is insufficient to carry out such detailed analysis. However, we have mentioned this analysis of stability as a perspective study. Additional information are given in lines (465-467)
3) There are many published works on the topic of "COVID-19". In the paper "How Does the Novel Coronavirus Interact with the Human ACE2 Enzyme? A Thermodynamic Answer", it is mentioned that compared with the old RBD, the Gibbs free energy of binding to hACE2 enzyme increases from 5.11 to 8.39 kcal/mol for the new RBD. Why is the free energy switch between closed and open state is 10-15 kcal/mol in this work?
Answer: We appreciate the opportunity to discuss our results further. The increment in free energy upon binding to ACE2 has been characterized in the suggested reference through molecular docking and MD simulations. Such study shows the comparison of the free energy for the binding of the RBD and the ACE2 receptor in SARS-CoV-2 and SARS-CoV-1 systems. In this regard, our MS tackles a rather different problem which is the calculation of the free energy associated with the transition from the closed state (3down) to one of the open conformations (e.g. 1up2down and 2up1down) in SARS-CoV-2 by looking at the structures, in the absence of the ACE2 receptor. This process is believed to happen spontaneously as a part of the dynamical equilibrium of the spike protein prior to ACE2 binding. Here, we show that if such a process happens it may cost an equivalent change in free energy, calculated by the change in protein conformation in water of 10 kcal/mol for the 3down to 1up2down transition. This energy is equivalent to saying that it might correspond to the rupture of 10 strong native contacts each with an energy in the range ~ 1kcal/mol. The energy scale for a native contact is known from simulation and experiment [16,43]. We have identified those 10 high-frequency contacts during the detachment of the RBD from the S2 domain mediated by the hinge domain. Our MS does not look into the mechanistic process of the rupture of those contacts that enables the transition closed to open, however, mechanical loading and the virus collision onto the cell could trigger such transition. Additional information that clarifies the message is given in lines (472-473) and list of 10 strong contacts can be found in Table S2 in the SM.
4) In the same work as mentioned in point 3), they authors have detected many active sides of forming hydrogen bonds. Are the obtained contacts match with these active sides?
Answer: In the above mentioned reference the list of residues in the RBD that establish several hydrogen bonds (HB) with the ACE2 during binding are A475, G485, N487, F490, G493, Y495, G496, G498, T500, N501, G502. Our study identifies the following two stabilizing, A475 and N487 , residues that overlap with residues important for ACE2 recognition. They may establish strong native contacts, possibly hydrogen bond type, within the RBD structure and mostly due to its polar character. We believe this information is out of scope as we did not analysis the ACE2 binding, so not changes are done.
5) It is known that protein structures are also dependent on the chosen all-atom force field. Therefore, the contact map may be also sensitive to the choice of the force field. Can the authors verify their force field choice? At least the criteria should be mentioned.
Answer: We thank the reviewer for this comment. We should point out that the Amber FF14SB force field is a very well-established protein force field from the Amber protein family of force fields. It emerges from the same line of force fields which include FF99SB and FF12SB, for example, which have been used and well-validated in the literature for many years. We include just two recent cases where the ff14SB force field was used for COVID-19 work, https://dx.doi.org/10.1126%2Fsciadv.abd0345 and https://doi.org/10.1021/acs.jcim.0c00575. Indeed, Woo et al. (https://dx.doi.org/10.1021%2Facs.jpcb.0c04553) use the GROMOS force field as their software of choices is Gromacs, but they indicate in their manuscript that of course other force fields can be used as well. Our purpose in this manuscript is not to perform a force-field comparison. In this regard any change is necessary.
6) Typos. For example, "as a pandemic..As" in page 2, "Delta Gs" instead of "\Delta Gs" in page 11. The authors use "Severe Acute Respiratory Syndrome" at one place but "severe acute respiratory syndrome" at another. Please use the same format.
Answer: The article has been proofread and several corrections have been done.

Reviewer 3 Report
The manuscript entitled “Characterization of Structural and Energetic Differences Between Conformations of the SARSCoV-2 Spike Protein” describes the structural play of spike protein and its interaction with ACE2. They discuss the protein structural stability based on the computational determination of the dynamic contact map and the energetic differences of various conformations. As generally expected, this new procedure gives a low frequency for contacts between structurally flexible parts of the protein, while highly stable structures exhibit high frequency.
The first process during the binding of spike protein to ACE2 believed to be the transition from down to up conformation of the spike protein. The dynamic contact map analysis and energy calculations reveal importance amino acid level details of this transition. Their analysis shows that the 1up2down conformation if more stable than the 2up1down conformation. The around 20 kcal/mol energy difference between these states probably show that spike protein adopts this conformation before interacting with ACE2. Even though there are no experimental validation of the computational results, this manuscript is suitable for the journal and will improve the readership.
There few small corrections noted below.
- Figure 2 and Figure 3 and not described in the text.
- Page 7: “Based on our analysis the 1up2down conformation shows less number of destabilizing residues than the 2up1down conformation. However, the stabilizing aa residues per protomer in up conformation seems to be comparable.”
This is an interesting observation that could benefit from an explanation using structure. This reviewer believes that adding a text section to Figure 2 could help to describe this.
- Page 9: “Such conformational transition emerges through the rearrangement of certain residues located at the bottom of the RBD, close to the hinge region (H519-S530) in chain B that change conformation and breaks 10 high-frequency contacts between hinge region and S2 in chain A in order to transit to the new position.”
It would be helpful if the hinge region can be highlighted in one of the pictures already in the manuscript.
- Page 10: “The PB scheme has several advantages, when tackling large biggish protein assemblies of different structures and trajectories provided, as it is the case of the coronavirus spike proteins PDBs. One of the key advantages is the computational feasibility when tackling protein assemblies in the order of 100 thousands of atoms in just an hour.”
These sentences could be described better.
Author Response
Reviewer 2
The manuscript entitled “Characterization of Structural and Energetic Differences Between Conformations of the SARSCoV-2 Spike Protein” describes the structural play of spike protein and its interaction with ACE2. They discuss the protein structural stability based on the computational determination of the dynamic contact map and the energetic differences of various conformations. As generally expected, this new procedure gives a low frequency for contacts between structurally flexible parts of the protein, while highly stable structures exhibit high frequency.
The first process during the binding of spike protein to ACE2 is believed to be the transition from down to up conformation of the spike protein. The dynamic contact map analysis and energy calculations reveal the importance of amino acid level details of this transition. Their analysis shows that the 1up2down conformation is more stable than the 2up1down conformation. The around 20 kcal/mol energy difference between these states probably show that spike protein adopts this conformation before interacting with ACE2. Even though there are no experimental validation of the computational results, this manuscript is suitable for the journal and will improve the readership.
There are a few small corrections noted below.
- Figure 2 and Figure 3 and not described in the text.
Answer: We appreciate the reviewer’s comment and the right discussion of the Figures 2 and 3 have been placed in the revised version. Additional information in track-changes mode regarding Figure 2 and 3 are given in lines (329-336)
2. Page 7: “Based on our analysis the 1up2down conformation shows less number of destabilizing residues than the 2up1down conformation. However, the stabilizing aa residues per protomer in up conformation seems to be comparable.”
This is an interesting observation that could benefit from an explanation using structure. This reviewer believes that adding a text section to Figure 2 could help to describe this.
Answer: We thank the reviewer for this comment. The description of Figure 2 has been expanded and described the structural consequences in the revised version. We believe the additional discussion regarding the chemistry of the interactions has also contributed to a better explanation of our work. Additional information regarding Figure 2 has ben given before and new entry of chemistry in lines (338-341).
- Page 9: “Such conformational transition emerges through the rearrangement of certain residues located at the bottom of the RBD, close to the hinge region (H519-S530) in chain B that change conformation and breaks 10 high-frequency contacts between hinge region and S2 in chain A in order to transit to the new position.”
It would be helpful if the hinge region can be highlighted in one of the pictures already in the manuscript.
Answer: Figure 1 has been modified accordingly in order to highly the RBD detachment from S2 mediated by the hinge region in the revised version. Also, we have added the list of those 10 contacts in the Table S2 in the SM.
- Page 10: “The PB scheme has several advantages, when tackling large biggish protein assemblies of different structures and trajectories provided, as it is the case of the coronavirus spike proteins PDBs. One of the key advantages is the computational feasibility when tackling protein assemblies in the order of 100 thousands of atoms in just an hour.”
These sentences could be described better.
Answer: Thanks for the comment. We have clarified the message in the revised version. See entry lines (429-434).

Reviewer 4 Report
Moreira and co-workers an interesting simulation and experimental study aimed at elucidating the structural and molecular characteristics of SARS-CoV-2 Spike Protein as well as the stability of protein-receptor interactions in order to find novel therapeutic approaches able to block S-protein interaction with ACE2 host’s receptor. Below are reported a few comments that will improve the quality of the manuscript:
1) The first part of the Introduction section is not focused to the main theme of the manuscript that is SARS-CoV-2. Please avoid to describe other infectious diseases, even that caused by CoVs;
2) In the methods section, the authors should indicate the temperatures in Celsius;
3) In the discussion section, the authors should claim the limits of their study. In addition, it was demonstrated that SARS-CoV-2 S-protein is able to interact with other receptors (e.g. nAchR) and it competes with other endogenous or exogenous ligands, like nicotine. Please, add some information about these aspects.
Author Response
Reviewer 3
Moreira and co-workers an interesting simulation and experimental study aimed at elucidating the structural and molecular characteristics of SARS-CoV-2 Spike Protein as well as the stability of protein-receptor interactions in order to find novel therapeutic approaches able to block S-protein interaction with ACE2 host’s receptor. Below are reported a few comments that will improve the quality of the manuscript:
1) The first part of the Introduction section is not focused on the main theme of the manuscript that is SARS-CoV-2. Please avoid to describe other infectious diseases, even that caused by CoVs;
Answer: The introduction has been modified accordingly. See change in lines (38-42)
2) In the methods section, the authors should indicate the temperatures in Celsius;
Answer: Additional information about temperature in °C degrees has been placed within parentheses wherever it makes relevance in the biomolecular community. However, the Kelvin unit is a reference in equilibration protocols in MD simulations. Changes are done in line 177 and 185
3) In the discussion section, the authors should claim the limits of their study. In addition, it was demonstrated that SARS-CoV-2 S-protein is able to interact with other receptors (e.g. nAchR) and it competes with other endogenous or exogenous ligands, like nicotine. Please, add some information about these aspects.
Answer: We have discussed the methodological limitations for the dCM and PB approaches regarding their applications for large protein complexes. The relevance with other ligands and receptors has been discussed in the revised version. New Ref. [9] about nAchR and discussion are given in lines [67-70]. Limitation os the methodology are given in lines (211-213) and (307-309)

Round 2
Reviewer 2 Report
The authors have addressed all my concerns.
Author Response
Thanks for the comment.